# Biomarkers of dairy fat intake, incident cardiovascular disease, and all-cause mortality: A cohort study, systematic review, and meta-analysis

Kathy Trieu[1☯], Saiuj Bhat[2☯], Zhaoli Dai[3,4], Karin Leander[5], Bruna Gigante[6], Frank Qian[7,8], Andres V. Ardisson Korat[9], Qi Sun[7,9], Xiong-Fei Pan[1,10,11], Federica Laguzzi[5], Tommy Cederholm[12], Ulf de Faire[5], Mai-Lis Hellénius[5], Jason H. Y. Wu[1‡], Ulf Risérus[12‡], Matti Marklund[1,12,13‡*]

1 The George Institute for Global Health, Faculty of Medicine, University of New South Wales, Sydney, Australia, 2 School of Medicine, The University of Western Australia, Crawley, Australia, 3 Centre for Health Systems and Safety Research, Faculty of Medicine, Health and Human Sciences, Macquarie University, Sydney, Australia, 4 Sydney Pharmacy School and the Charles Perkins Centre, Faculty of Medicine and Health Sciences, University of Sydney, Sydney, Australia, 5 Unit of Cardiovascular and Nutritional Epidemiology, Institute of Environmental Medicine, Karolinska Institutet, Stockholm, Sweden, 6 Cardiovascular Medicine Unit, Department of Medicine, Karolinska Institutet, Stockholm, Sweden, 7 Department of Nutrition, Harvard T.H. Chan School of Public Health, Boston, Massachusetts, United States of America, 8 Department of Medicine, Beth Israel Deaconess Medical Center, Harvard Medical School, Boston, Massachusetts, United States of America, 9 Channing Division of Network Medicine, Department of Medicine, Brigham and Women's Hospital and Harvard Medical School, Boston, Massachusetts, United States of America, 10 Division of Epidemiology, Department of Medicine, Vanderbilt University Medical Center, Nashville, Tennessee, United States of America, 11 Department of Epidemiology and Biostatistics, School of Public Health, Tongji Medical College, Huazhong University of Science and Technology, Wuhan, Hubei, China, 12 Department of Public Health and Caring Sciences, Clinical Nutrition and Metabolism, Uppsala University, Uppsala, Sweden, 13 Department of Epidemiology, Johns Hopkins Bloomberg School of Public Health, Baltimore, Maryland, United States of America

☯ These authors contributed equally to this work.
‡ These authors are joint senior authors on this work.
* mmarklund@georgeinstitute.org.au

## Abstract

### Background

We aimed to investigate the association of serum pentadecanoic acid (15:0), a biomarker of dairy fat intake, with incident cardiovascular disease (CVD) and all-cause mortality in a Swedish cohort study. We also systematically reviewed studies of the association of dairy fat biomarkers (circulating or adipose tissue levels of 15:0, heptadecanoic acid [17:0], and *trans*-palmitoleic acid [*t*16:1n-7]) with CVD outcomes or all-cause mortality.

### Methods and findings

We measured 15:0 in serum cholesterol esters at baseline in 4,150 Swedish adults (51% female, median age 60.5 years). During a median follow-up of 16.6 years, 578 incident CVD events and 676 deaths were identified using Swedish registers. In multivariable-adjusted models, higher 15:0 was associated with lower incident CVD risk in a linear dose–response

**Data Availability Statement:** Data from the systematic review are within the manuscript and Supporting information files. Due to legal restrictions, some access restrictions apply to the

remaining data underlying the findings. Requests could be sent to Karolinska Institutet, Contact details: Karolinska Institutet, 171 77 Stockholm, Sweden, Telephone: +46 8 524 800, https://ki.se/en/about/how-can-we-help-you.

**Funding:** This work was supported by funds to UdF from Stockholm County Council (Stockholms Läns Landsting), Swedish Heart and Lung-Foundation (Hjärt-Lungfonden), the Swedish Research Council (Vetenskapsrådet). UR was supported by Swedish Heart and Lung-Foundation (Hjärt-Lungfonden). KT, JW and MM are researchers within a National Health and Medical Research Council of Australia (NHMRC) Centre for Research Excellence in reducing salt intake using food policy interventions (APP1117300). KT was supported by an Early Career Fellowship (APP1161597) from the NHMRC and a Postdoctoral Fellowship (Award ID 102140) from the National Heart Foundation of Australia. ZD was supported by a NHMRC project grant (APP1139997). JW was supported by a University of New South Wales Scientia Fellowship. AVAK was supported by National Institutes of Health (T32 CA009001). The funder had no role in the study design, data collection and analysis, decision to publish, or preparation of the manuscript.

**Competing interests:** The authors have declared that no competing interests exist.

**Abbreviations:** CHD, coronary heart disease; CI, confidence interval; CVD, cardiovascular disease; HR, hazard ratio; IQR, interquintile range; LDL, low-density lipoprotein; NOS, Newcastle–Ottawa Scale; PD, percentile difference; PUFA, polyunsaturated fatty acid; RR, relative risk.

manner (hazard ratio 0.75 per interquintile range; 95% confidence interval 0.61, 0.93, $P = 0.009$) and nonlinearly with all-cause mortality (P for nonlinearity = 0.03), with a nadir of mortality risk around median 15:0. In meta-analyses including our Swedish cohort and 17 cohort, case–cohort, or nested case–control studies, higher 15:0 and 17:0 but not $t$16:1n-7 were inversely associated with total CVD, with the relative risk of highest versus lowest tertile being 0.88 (0.78, 0.99), 0.86 (0.79, 0.93), and 1.01 (0.91, 1.12), respectively. Dairy fat biomarkers were not associated with all-cause mortality in meta-analyses, although there were ≤3 studies for each biomarker. Study limitations include the inability of the biomarkers to distinguish different types of dairy foods and that most studies in the meta-analyses (including our novel cohort study) only assessed biomarkers at baseline, which may increase the risk of misclassification of exposure levels.

## Conclusions

In a meta-analysis of 18 observational studies including our new cohort study, higher levels of 15:0 and 17:0 were associated with lower CVD risk. Our findings support the need for clinical and experimental studies to elucidate the causality of these relationships and relevant biological mechanisms.

## Author summary

### Why was this study done?

- Many dietary guidelines recommend limiting dairy fat consumption in order to lower saturated fat intake and cardiovascular disease (CVD) risk.

- However, increasing evidence suggests that the health impact of dairy foods is more dependent on the type (e.g., cheese, yoghurt, milk, and butter) rather than the fat content, which has raised doubts if avoidance of dairy fats is beneficial for cardiovascular health.

- Dairy foods are a major source of nutrients, and their consumption is increasing worldwide; thus, it is important to advance our understanding of the impact of dairy fat on CVD risk.

### What did the researchers do and find?

- We measured dairy fat consumption using an objective biomarker, serum pentadecanoic acid (15:0), in 4,150 Swedish 60-year-olds and collected information about CVD events and deaths during a median follow-up of 16.6 years.

- When we accounted for known risk factors including demographics, lifestyle, and disease prevalence, the CVD risk was lowest for those with high levels of the dairy fat biomarker 15:0, while those with biomarker levels around the median had the lowest risk of all-cause mortality.

- We also conducted a systematic review and meta-analysis, and the combined evidence from 18 studies also showed higher levels of 2 dairy fat biomarkers (15:0 and heptadecanoic acid 17:0) were linked with lower risk of CVD, but not with all-cause mortality.

### What do these findings mean?

- The findings from our study using fatty acid biomarkers suggest that higher intake of dairy fat were associated with lower CVD risk in diverse populations including Sweden (a country with high dairy intake), though more trials are needed to understand if and how dairy foods protect cardiovascular health.

## Introduction

Cardiovascular disease (CVD) is the leading cause of mortality worldwide, responsible for almost 1 in every 3 deaths [1]. While in past decades guidelines generally suggested the avoidance of dietary fats for cardiovascular health, there is now growing evidence that the type and dietary source of fat may be more important for CVD risk than the total amount [2,3]. In particular, there is emerging evidence regarding the role of dairy fats and CVD. While increased intake of saturated fat from dairy is expected to increase low-density lipoprotein (LDL) cholesterol, recent human clinical studies found that such effects differ depending on the type of dairy products as well as the processing methods [4,5]. Long-term observational studies have found no association between total dairy consumption and risk of CVD, with differences in association observed for the type of dairy product rather than the amount of fat in dairy products (e.g., regular versus reduced fat dairy products) [4,6,7]. For example, fermented dairy products, such as cheese and yoghurt, may be more protective than milk and butter [8,9]. Such findings have generated debate as to whether dietary or clinical guidelines based predominantly on considerations of the saturated fat content of dairy foods are appropriate [10].

Studies have traditionally relied upon self-reported measures of dairy fat intake that are subject to recall bias and may be limited in capturing the plethora of dairy-containing foods or by systematic errors in food composition databases [11]. To overcome these limitations, fatty acid composition in tissues or the circulation are increasingly being utilised as biomarkers of dietary fat [12]. Two odd-chain saturated fatty acids, pentadecanoic acid (15:0) and heptadecanoic acid (17:0), and one *trans*-fatty acid, *trans*-palmitoleic acid (*t*16:1n-7), are increasingly used as biomarkers of dairy fat intake because they are mainly found in ruminant foods such as milk and are not strongly influenced by genetic variation [12,13]. Thus, their levels correlate with dairy fat consumption assessed through weighed diet records and 24-hour dietary recalls and change in accordance with dairy food intake in randomised controlled trials [12,14,15].

Since dairy foods are a major source of nutrients and increasingly consumed globally [16], it is crucial to have a better understanding of the impact of dairy fat intake on CVD risk. Within this context, we aimed to investigate the association of serum pentadecanoic acid (15:0) with incident CVD and all-cause mortality in a Swedish population-based cohort and incorporated these data in a systematic review of prospective studies evaluating the associations of circulating or adipose tissue dairy fat biomarkers (15:0, 17:0, and *t*16:1n-7) with incident CVD or all-cause mortality.

## Methods

### Cohort study

**Study design and population.** The Stockholm Cohort of 60-year-olds (60YO) has been previously described [17]. One-third of men and women aged 60 between 1 July 1997 and 30 June 1998 residing in Stockholm County ($n$ = 5,460) were randomly selected from the population register and invited to participate in the study. Of these, 4,232 (78%) agreed to participate (52% women) and provided informed consent. The participants underwent a health screening, including blood sampling and completion of an extensive questionnaire (S1 Text). For the current analysis, 4,150 participants that had fasting blood samples collected at baseline between 1997 and 1999, and follow-up information until 31 December 2014 were included. The study was approved by the Ethics Committee at Karolinska Institutet, and all participants provided their informed consent to participate.

**Exposure assessment.** Blood samples were collected from participants after an overnight fast, and the serum samples were stored at −80 ˚C. Fatty acid composition in serum cholesterol esters was measured by gas chromatography as described previously [18]. Briefly, serum cholesteryl esters were methylated, extracted in petroleum ether, evaporated under nitrogen, and then redissolved in hexane before analysis by gas chromatography using a 30-m glass capillary column coated with Thermo TR-FRAME; an Agilent Technologies system consisting of model GLAC 6890N, an autosampler 7683, and Agilent ChemStation; with a programmed temperature of between 150 ˚C to 260 ˚C. Thirteen different fatty acids were quantified, and the proportion of each was expressed as a percentage of all fatty acids measured. The intra-assay and inter-assay coefficient of variations for 15:0 acid were 3.6% and 7.6%, respectively. As previously described, serum 15:0 was associated with self-reported dairy intake in 60YO (Fig A in S1 File) [19].

**Outcome assessment.** The primary outcomes were incident CVD and all-cause mortality retrieved from the Swedish Hospital Discharge and Cause of Death Registers. Incident CVD was defined as first-time CVD events including fatal and nonfatal myocardial infarction, fatal and nonfatal ischaemic stroke, and hospitalisation resulting from angina pectoris (International Classification of Disease, 10th Revision codes: I20, I21, I25, I46, and I63 to I66) [17]. Secondary outcomes included CVD mortality, defined as deaths caused by CVD, incident coronary heart disease (CHD), and incident ischaemic stroke. CHD and ischaemic stroke were mutually exclusive events such that participants were censored after their first CVD event.

**Statistical analysis.** Analytic methods were prespecified in a protocol (S1 Protocol). Participants with CVD at baseline were excluded from the analyses of incident CVD, CHD, and ischaemic stroke. Cox proportional hazard models were used to estimate hazard ratios (HRs) and 95% confidence intervals (CIs) for the association between serum 15:0 with primary and secondary outcomes. Differences in time to first CVD event or death by serum 15:0 levels were estimated using Laplace regression [20]. During follow-up, around 15% of participants died, and a similar number of persons experienced a CVD event. Hence, we estimated the 15th percentile difference (PD) defined as the difference in time (months) by which 15% of exposed versus unexposed had died or experienced an incident CVD event. Evaluation of PD at percentiles lower than the 15th (i.e., first to 14th) provided similar results. Three models were evaluated: (1) crude, without adjustments; (2) age- and sex-adjusted; and (3) multivariable-adjusted including age, sex, BMI, smoking, physical activity, education, alcohol intake, diabetes, drug-treated hypertension, and drug-treated hypercholesterolaemia as covariates (S1 Text). For analyses of all-cause or CVD mortality, the multivariable-adjusted model also included prevalent CVD as a covariate. Multiple imputations ($n$ = 20) were conducted to account for missing

covariates. Less than 3% of the study population had missing values for $\geq 1$ covariate, and the frequency of missing values in each covariate was <1%. Serum 15:0 was assessed as a continuous variable (per interquintile range (IQR), defined as the range between the 90th and 10th percentiles) or a categorical variable (quartiles). There was no violation of the proportional hazard assumption based on visual examination of Schoenfield residuals. Restricted cubic splines were used to evaluate potential nonlinear associations. We explored the associations between serum 15:0 and outcomes in subgroup analyses stratified by sex, BMI, and serum n-3 polyunsaturated fatty acid (PUFA) subgroups (< median versus $\geq$ median).

We conducted sensitivity analyses by (1) adjusting for self-reported dietary habits (vegetable, fruit and berries, lean fish, oily fish, and processed meat intake) (which was not prespecified); (2) excluding participants with prevalent CVD also from analyses of all-cause mortality (in line with analyses of CVD outcomes); (3) restricting analyses to the first 10 years of follow-up to minimise misclassifications attributable to exposure changes over time; and (4) excluding cases in the first 2 years of follow-up to avoid reverse causation because of undetected disease or presence of risk factors at baseline.

This study is reported as per the Strengthening the Reporting of Observational Studies in Epidemiology (STROBE) guideline (S1 Checklist).

## Systematic review and meta-analysis

**Search strategy.** A systematic literature search up to 27 June 2021 was conducted in Medline, Embase, Scopus, Web of Science, and CENTRAL databases using a search strategy detailed in S2 Text [21]. The systematic review followed the PRISMA guidelines (S2 Checklist) and was registered on PROSPERO [CRD42020162551].

**Study selection and data extraction.** The studies eligible for inclusion were prospective observational human studies that examined the association between circulating or adipose tissue levels of 15:0, 17:0, or $t$16:1n-7 at baseline and risk of CVD events or mortality during follow-up. Prospective cohort, case–cohort, and nested case–control studies were included. Studies were excluded if they had a retrospective or cross-sectional design, standard errors were missing or could not be calculated, all participants had CVD at baseline, or they did not adjust for confounders. Two reviewers (SB and ZD) independently screened the studies for eligibility, extracted data, and assessed the quality of studies using the Newcastle–Ottawa Scale (NOS) [22]. Disagreements were resolved by consensus or by involvement of a third reviewer (JW).

**Meta-analysis.** Pooled associations of 15:0, 17:0, and $t$16:1n-7 with CVD outcomes and all-cause mortality were estimated using random effects meta-analysis. The primary outcomes included total CVD and all-cause mortality. CVD, CHD, stroke (incidence and mortality), as well as incident heart failure were evaluated in secondary analyses. For the analysis of total CVD, the effect size estimate for each study was selected in the following order: CVD incidence > CVD mortality > CHD, stroke, or heart failure incidence > CHD or stroke mortality > other CVD outcome. Risk estimates (HR, odds ratio, or relative risk (RR)) for each study were transformed to allow consistent comparisons between the top and bottom tertile of fatty acid distributions (S2 Text). We included multiple risk estimates from the same cohort if they were derived from separate nested case–control studies. In addition to the quantile analysis, dairy fat biomarkers were evaluated as continuous variables (per 1 SD increase) in a subset of studies with relevant information available or retrieved from study authors. For studies that provided estimates of dairy fat biomarkers in more than one biological tissue, one effect estimate was selected using the following hierarchy to preference lipid compartments that reflect longer-term fatty acid intake: adipose tissue > erythrocyte or plasma

phospholipids > cholesterol esters > total plasma. The $I^2$ and Q statistics were used to assess heterogeneity of included studies. Publication bias was assessed by visual inspection of funnel plots and statistically using Egger's and Begg's tests. Stratified meta-analyses were performed on subgroups defined by age, sex, follow-up duration, and geographic region. We repeated the meta-analysis using a fixed effects models in a sensitivity analysis. All statistical tests were performed with STATA 15 (Stata Corp, College Station, TX), two-sided, and a $P < 0.05$ was considered statistically significant.

## Results

### 60YO cohort study—Serum 15:0 and incident CVD and mortality

At baseline, median age was 60.5 years, 51% ($n$ = 2,133) were women, median BMI was 26 kg/m$^2$, 8% had prevalent type 2 diabetes, and 9% had prevalent CVD (Table 1 and Table A in S1 File). During a median follow-up duration of 16.6 years, 578 incident CVD events (386 CHD events and 192 ischaemic strokes) occurred over 55,832 person-years and 676 deaths (198 due to CVD) occurred over 64,605 person-years.

**Incident CVD.** In multivariable-adjusted models, higher serum 15:0 was associated with lower incident CVD in a linear dose–response manner (HR 0.75 per IQR; 95% CI 0.61, 0.93,

**Table 1. 60YO study population characteristics at baseline[1].**

|  | Women | Men | Total |
|---|---|---|---|
| N (%) | 2,133 (51) | 2,017 (49) | 4,150 (100) |
| Age, y | 60.4 (60.4, 60.7) | 60.5 (60.4, 60.7) | 60.5 (60.4, 60.7) |
| BMI, kg/m$^2$ | 26.0 (21.6, 32.9) | 26.6 (22.6, 31.8) | 26.3 (22.1, 32.2) |
| Alcohol intake, g/d | 4.9 (0.0, 20.3) | 13.9 (1.3, 40.9) | 8.5 (0.6, 32.8) |
| Serum cholesterol ester FA, % of total FA |  |  |  |
| Pentadecanoic acid | 0.21 (0.17, 0.27) | 0.22 (0.17, 0.28) | 0.22 (0.17, 0.28) |
| Long-chain n-3 PUFA | 2.78 (1.89, 4.39) | 2.73 (1.76, 4.47) | 2.75 (1.81, 4.42) |
| Physical activity, n (%) |  |  |  |
| Sedentary | 245 (11) | 208 (10) | 453 (11) |
| Light exercise | 1,252 (59) | 1,058 (52) | 2,310 (56) |
| Moderate exercise | 431 (20) | 485 (24) | 916 (22) |
| Regular exercise | 120 (6) | 175 (9) | 295 (7) |
| Smoking, n (%) |  |  |  |
| Never | 942 (44) | 638 (32) | 1,580 (38) |
| Former | 651 (31) | 892 (44) | 1,543 (37) |
| Current | 454 (21) | 397 (20) | 851 (21) |
| Disease prevalence, n (%) |  |  |  |
| Type 2 diabetes | 113 (5) | 199 (10) | 312 (8) |
| CVD | 139 (7) | 226 (11) | 365 (9) |
| Drug-treated hypertension | 372 (17) | 423 (21) | 795 (19) |
| Drug-treated hyperlipidaemia | 79 (4) | 135 (7) | 214 (5) |
| Education, n (%) |  |  |  |
| Primary school (≤9 y) | 625 (29) | 521 (26) | 1,146 (28) |
| Secondary school (>9 y, ≤12 y) | 875 (41) | 870 (43) | 1,745 (42) |
| University or college (>12 y) | 557 (26) | 544 (27) | 1,101 (27) |

[1]Values are median (10th and 90th percentiles) or n (%).

CVD, cardiovascular disease; FA, fatty acid; PUFAs, polyunsaturated fatty acids.

**Table 2. HRs and 15th PDs of incident CVD and all-cause mortality per IQR of serum pentadecanoic acid (15:0) in the 60YO study[1].**

| Outcome | | Model[2] | | P_linear[3] | P_nonlinear[4] |
|---|---|---|---|---|---|
| Incident CVD[5] | Cases/person-years | | 578/55,832 | | |
| | HR (95% CI)[6] | 1 | 0.73 (0.59, 0.89) | 0.003 | 0.50 |
| | | 2 | 0.64 (0.52, 0.80) | <0.001 | 0.52 |
| | | 3 | 0.75 (0.61, 0.93) | 0.009 | 0.98 |
| | PD (95% CI)[7], months | 1 | 41.1 (14.7, 67.6) | 0.002 | 0.13 |
| | | 2 | 48.4 (21.9, 74.9) | <0.001 | 0.78 |
| | | 3 | 27.0 (6.1, 48.0) | 0.01 | 0.82 |
| All-cause mortality | Cases/person-years | | 676/64,605 | | |
| | HR (95% CI)[6] | 1 | 0.74 (0.60, 0.92) | 0.006 | <0.001 |
| | | 2 | 0.72 (0.58, 0.89) | 0.002 | <0.001 |
| | | 3 | 0.91 (0.74, 1.12) | 0.38 | 0.03 |
| | PD (95% CI)[7], months | 1 | 33.7 (13.5, 53.8) | 0.001 | <0.001 |
| | | 2 | 26.6 (5.2, 47.9) | 0.01 | <0.001 |
| | | 3 | 4.5 (−13.7, 22.6) | 0.63 | 0.24 |

[1]CI, confidence interval; CVD, cardiovascular disease; HR, hazard ratio; PD, 15th survival percentile difference.

[2]Model 1 includes serum 15:0 as the only covariate and was thus used to assess crude associations. Model 2 included adjustments for age and sex. Model 3 was further adjusted for BMI, alcohol intake, smoking habits, physical activity, education, and prevalent hypertension, hyperlipidaemia, type 2 diabetes, and (for evaluation of all-cause mortality) CVD.

[3]Linear associations were evaluated per IQRs (i.e., midpoints of the first and fifth quintiles) of biomarker 15:0.

[4]Nonlinear trends were evaluated using restricted cubic splines (knots at 10th, 50th, and 90th percentiles).

[5]Participants with prevalent CVD at baseline were excluded from analyses on incident CVD.

[6]HRs were estimated using Cox proportional hazard models.

[7]Laplace regression was used to model 15th percentile survival.

$P = 0.009$) (Table 2, Fig 1, and Table B in S1 File showing HR per SD and % of total fatty acids). The time by which 15% experienced an incident CVD event increased by 27 months (95% CI: 6, 48) per IQR (Table 2). Evaluating quartiles of serum 15:0, CVD risk was lower at higher 15:0 levels (P-trend = 0.016), with HR of the top versus bottom quartile of 0.76 (95% CI: 0.59, 0.97) after adjustment for confounders (Table 3). In secondary analyses, higher serum 15:0 was significantly associated with lower risk of CHD (HR 0.70 per IQR; 95% CI: 0.54, 0.91) but not ischaemic stroke (HR 0.87 per IQR; 95% CI: 0.61, 1.25) (Table C in S1 File).

**Mortality.** In the multivariable-adjusted model, there was no significant linear association of serum 15:0 with all-cause mortality ($P = 0.38$) (Table 2). However, results from the spline analyses suggested a nonlinear association (P for nonlinearity = 0.03) (Table 2), with a nadir of mortality risk around median 15:0 (i.e., 0.22% of total fatty acids) (Fig 2). Evaluation of serum 15:0 quartiles further supported the findings of a nonlinear association, with a 22% lower mortality risk (HR 0.78; 95% CI: 0.62, 0.98) in the third versus first quartile (Table 3). This lower mortality risk translated into a longer survival; the time by which 15% died was 23 months later (95% CI: 1, 45) in the third versus first quartile. Serum 15:0 was not significantly associated with CVD mortality after adjustment for potential confounders (Table C in S1 File).

**Stratified and sensitivity analyses.** Associations of serum 15:0 and CVD or mortality outcomes did not differ by sex, BMI, or serum n-3 PUFA levels (Table D in S1 File). Results from sensitivity analyses that adjusted for self-reported dietary habits, excluded early cases (≤2 years after baseline), censored follow-up at 10-years, or excluded individuals with prevalent CVD (in the mortality analyses) did not alter our findings (Table E in S1 File).

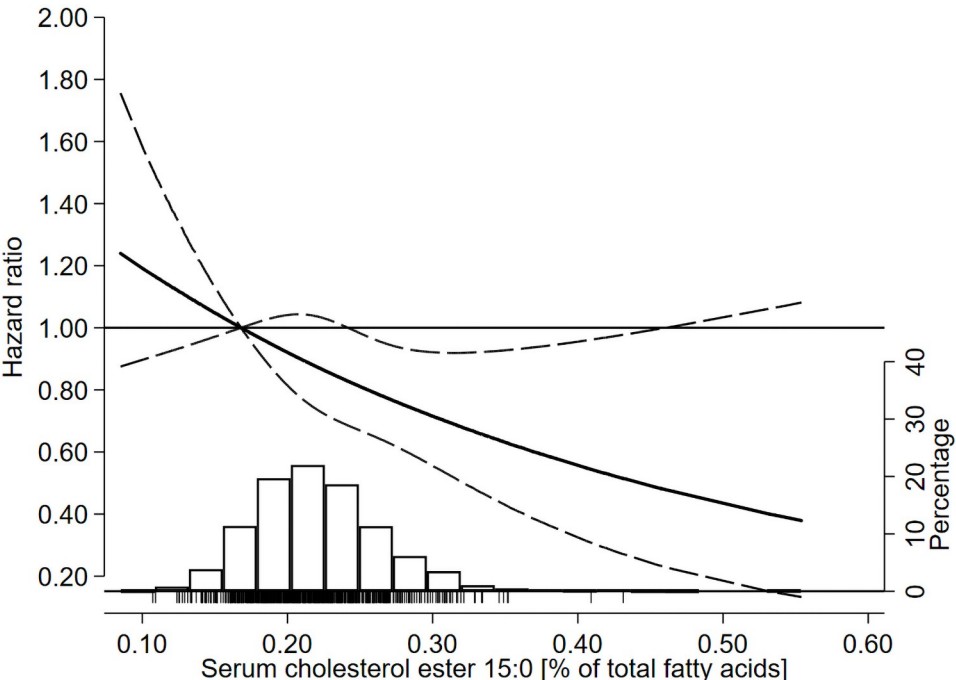

**Fig 1. HRs of incident CVD as a function of serum pentadecanoic acid (15:0) in the 60YO study.** Data were fitted using Cox regression models adjusted for baseline age, sex, BMI, alcohol intake, smoking habits, physical activity, education, and prevalent hypertension, hyperlipidaemia, and type 2 diabetes. Dashed lines represent 95% confidence limits. The reference value of serum 15:0 is the 10th percentile (i.e., 0.17% of total fatty acids). The histogram shows the distribution of serum 15:0 in the cohort, and the tick marks under the histogram indicate serum 15:0 levels of individuals who experienced an incident CVD event during follow-up. CVD, cardiovascular disease; HR, hazard ratio.

## Systematic review and meta-analysis

A systematic review of the literature identified 18 studies, including the 60YO, that met the inclusion criteria (Fig 3) [23–41]. The characteristics of the studies and their quality assessment based on NOS are presented in Tables F and G in S1 File. All except one study were considered good-quality studies scoring a total NOS score of between 6 and 9 out of 9, and one study was considered fair quality (Table G in S1 File). The 18 studies together included 42,736 participants (although actual numbers differed for each dairy fat biomarker), and 11,950 total CVD cases were analysed in studies evaluating 15:0; 9,009 in studies evaluating 17:0; and 3,477 in studies evaluating $t$16:1n-7.

In pooled analyses evaluating high versus low levels of dairy fat biomarkers, 15:0 and 17:0 (Figs 4 and 5), but not $t$16:1n-7 (Table 4), were inversely associated with total CVD. The RR estimates (95% CI; n studies) of total CVD for the top versus bottom tertiles of 15:0, 17:0, and $t$16:1n-7 were 0.88 (0.78 to 0.99; $n = 17$), 0.86 (0.79 to 0.93; $n = 12$), and 1.01 (0.91 to 1.12; $n = 6$), respectively. The highest versus lowest tertiles of 17:0, but not 15:0 and $t$16:1n-7, were significantly associated with lower CHD and stroke risks (Table 4). In the studies allowing evaluation of continuous exposure, each 1 SD increase of 15:0 and 17:0 was inversely associated with total CVD, with RR 0.93 (95% CI 0.86 to 1.00; $n = 12$) for 15:0 and 0.93 (0.88 to 0.98; $n = 9$) for 17:0 (Table H in S1 File). Higher versus lower levels of dairy fat biomarkers were not significantly associated with all-cause mortality (Figs 4 and 5 and Table 4).

There was no statistical evidence that associations of dairy fat biomarkers and CVD risk were modified by age, sex, follow-up duration, or region (Europe versus United States)

**Table 3.  HRs and 15th PDs of incident CVD and all-cause mortality by quartile of serum pentadecanoic acid (15:0) in the 60YO study[1].**

| Outcome | | Model[2] | Quartile | | | | P_trend[3] |
|---|---|---|---|---|---|---|---|
| | | | Q1 | Q2 | Q3 | Q4 | |
| | Serum 15:0; median (min-max) | | 0.17 (0.09, 0.19) | 0.20 (0.19, 0.22) | 0.23 (0.22, 0.25) | 0.27 (0.25, 0.55) | |
| Incident CVD[4] | Cases (person-years) | | 168 (13,485) | 149 (13,812) | 132 (14,248) | 129 (14,287) | |
| | HR (95% CI)[5] | Model 1 | Ref | 0.86 (0.69, 1.08) | 0.74 (0.59, 0.93) | 0.72 (0.57, 0.91) | 0.003 |
| | | Model 2 | Ref | 0.85 (0.68, 1.06) | 0.69 (0.55, 0.86) | 0.63 (0.50, 0.80) | <0.001 |
| | | Model 3 | Ref | 0.92 (0.73, 1.16) | 0.79 (0.62, 1.00) | 0.76 (0.59, 0.97) | 0.016 |
| | PD (95% CI), months[6] | Model 1 | Ref | 28.1 (−3.2, 59.3) | 46.1 (11.3, 80.8) | 48.5 (15.4, 81.6) | <0.001 |
| | | Model 2 | Ref | 16.2 (−9.7, 42.2) | 36.6 (10.8, 62.3) | 50.8 (30.0, 71.7) | <0.001 |
| | | Model 3 | Ref | 9.8 (−14.1, 33.7) | 19.3 (−6.8, 45.4) | 29.1 (1.9, 56.3) | 0.023 |
| All-cause mortality | Cases (person-years) | | 195 (15,813) | 191 (15,991) | 134 (16,440) | 156 (16,361) | |
| | HR (95% CI) | Model 1 | Ref | 0.97 (0.79, 1.18) | 0.65 (0.52, 0.82) | 0.77 (0.62, 0.95) | 0.002 |
| | | Model 2 | Ref | 0.97 (0.80, 1.19) | 0.65 (0.52, 0.81) | 0.73 (0.59, 0.91) | <0.001 |
| | | Model 3 | Ref | 1.11 (0.90, 1.36) | 0.78 (0.62, 0.98) | 0.96 (0.77, 1.21) | 0.38 |
| | PD (95% CI), months | Model 1 | Ref | 11.4 (−16.5, 39.4) | 49.0 (24.2, 73.8) | 36.1 (6.8, 65.4) | <0.001 |
| | | Model 2 | Ref | −0.5 (−19.8, 18.8) | 38.2 (16.7, 59.8) | 26.6 (3.9, 49.2) | 0.005 |
| | | Model 3 | Ref | −0.7 (−23.0, 21.7) | 22.9 (1.0, 44.8) | 7.1 (−14.7, 28.8) | 0.26 |

[1]CI, confidence interval; CVD, cardiovascular disease; HR, hazard ratio; Q, quartile; PD, 15th survival percentile difference.

[2]Model 1 includes serum 15:0 as the only covariate and was thus used to assess crude associations. Model 2 included adjustments for age and sex. Model 3 was further adjusted for BMI, alcohol intake, smoking habits, physical activity, education, and prevalent hypertension, hyperlipidaemia, type 2 diabetes, and (for evaluation of all-cause mortality) CVD.

[3]Linear trend (P_trend) across quartiles of fatty acid biomarker was assessed by using linear trends with quartile medians (expressed as % of total fatty acids) as exposure.

[4]Participants with prevalent CVD at baseline were excluded from analyses on incident CVD.

[5]HR were estimated using Cox proportional hazard models.

[6]Laplace regression was used to model 15th percentile survival.

(Table I in S1 File). However, there was evidence of heterogeneity by age in association of 15:0 and total CVD ($P = 0.020$). There was no evidence of publication bias on visual inspection of the funnel plots (Fig B in S1 File) and by Begg's and Egger's tests.

## Discussion

In a population-based Swedish cohort study (i.e., 60YO), higher circulating levels of 15:0, a biomarker of dairy fat intake, were inversely associated with incident CVD. These findings were supported by our systematic review, which represents the most up-to-date and comprehensive synthesis of the evidence on the relation between dairy fat biomarkers, CVD, and mortality. Overall, higher levels of both odd-chain dairy fat biomarkers 15:0 and 17:0 were associated with 12% to 14% lower risk of CVD, comparing top versus bottom thirds of biomarker levels. Conversely, t16:1n-7 was not related to the risk of CVD. Our meta-analysis results from 3 available studies examining dairy fat biomarkers in relation to all-cause mortality show no clear associations.

Compared to their saturated even-chain fatty acid counterparts, there has been relatively little research into how odd-chain fatty acids might influence cardiovascular risk factors. A recent animal experimental study demonstrated that daily oral supplementation of 15:0 decreased proinflammatory states in obese mice with metabolic syndrome and also lowered total cholesterol [42]. However, this requires replication in human studies. Although the direct metabolic effects of 15:0 and 17:0 are unclear, there is good evidence to suggest that the levels

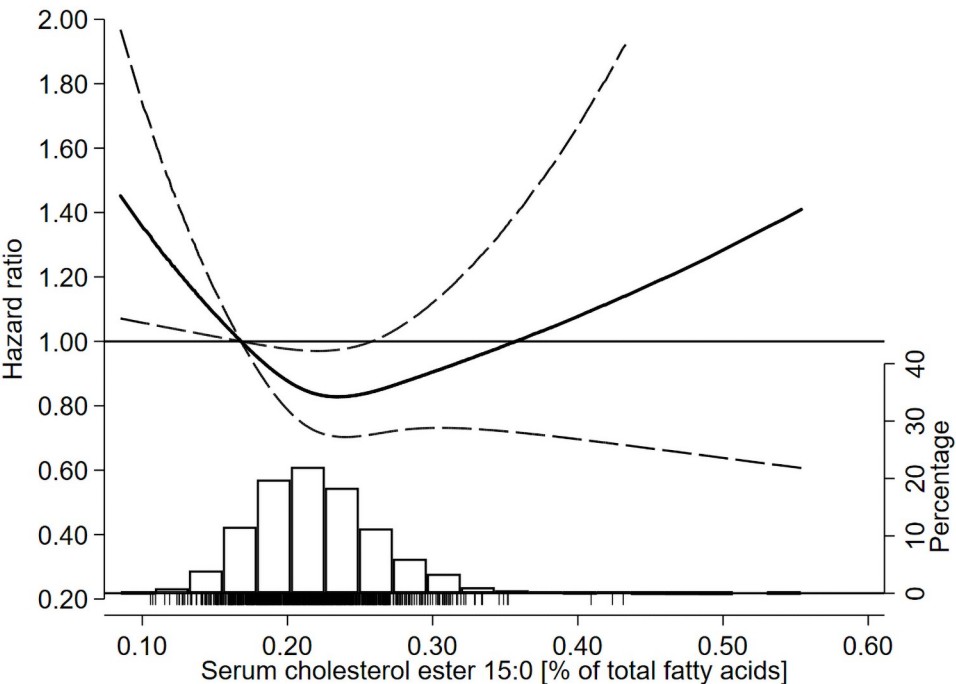

**Fig 2. HRs of all-cause mortality as a function of serum pentadecanoic acid (15:0) in the 60YO study.** Data were fitted using Cox regression models adjusted for baseline age, sex, BMI, alcohol intake, smoking habits, physical activity, education, and prevalent hypertension, hyperlipidaemia, type 2 diabetes, and CVD. Dashed lines represent 95% confidence limits. The reference value of serum 15:0 is the 10th percentile (i.e., 0.17% of total fatty acids). The histogram shows the distribution of serum 15:0 in the cohort and the tick marks under the histogram indicate serum 15:0 levels of individuals who died during follow-up. CVD, cardiovascular disease; HR, hazard ratio.

of odd-chain fatty acids may reflect intake of other constituents or nutrients in dairy fat or dairy fat–rich foods that have potential cardiometabolic benefits [12]. For instance, cheese is a major dietary source of vitamin K. Vitamin K may influence CVD risk through vitamin K–dependent proteins and reductions in vascular calcification, although evidence from prospective studies for cardiovascular benefit remains limited and conflicting [43]. Probiotics in dairy foods (such as yoghurt and fermented milk) and their interaction with the human gut microbiota may also confer cardiometabolic benefits [2]. Together, such potential cardioprotective components of dairy foods may partly explain our findings, and our findings are also consistent with prior meta-analyses that show self-reported dairy intake were associated with reduced risk of total CVD by 10% to 12% [44,45]. Given the long-standing and prevalent dietary guidance to consume low-fat dairy products [10], our results highlight the importance of additional animal-experimental and clinical research into the biologic mechanisms whereby odd-chain dairy fatty acids may influence and prevent CVD.

Dairy and dairy product consumption in Sweden is among the highest worldwide, and their health benefits in the Nordic diet have long been debated [46,47]. While most of the prior prospective studies have focused on cardiovascular outcomes, recent large Swedish studies found higher self-reported intake of nonfermented milk to be positively associated with all-cause mortality [48–50], which was in contrast with meta-analyses of evidence from other countries that found null associations [7,51]. Using an objective measure of dairy fat intake, our findings from the 60YO study suggest a nonlinear association between 15:0 and all-cause mortality. Importantly, even at very high levels of 15:0, there was no significant association

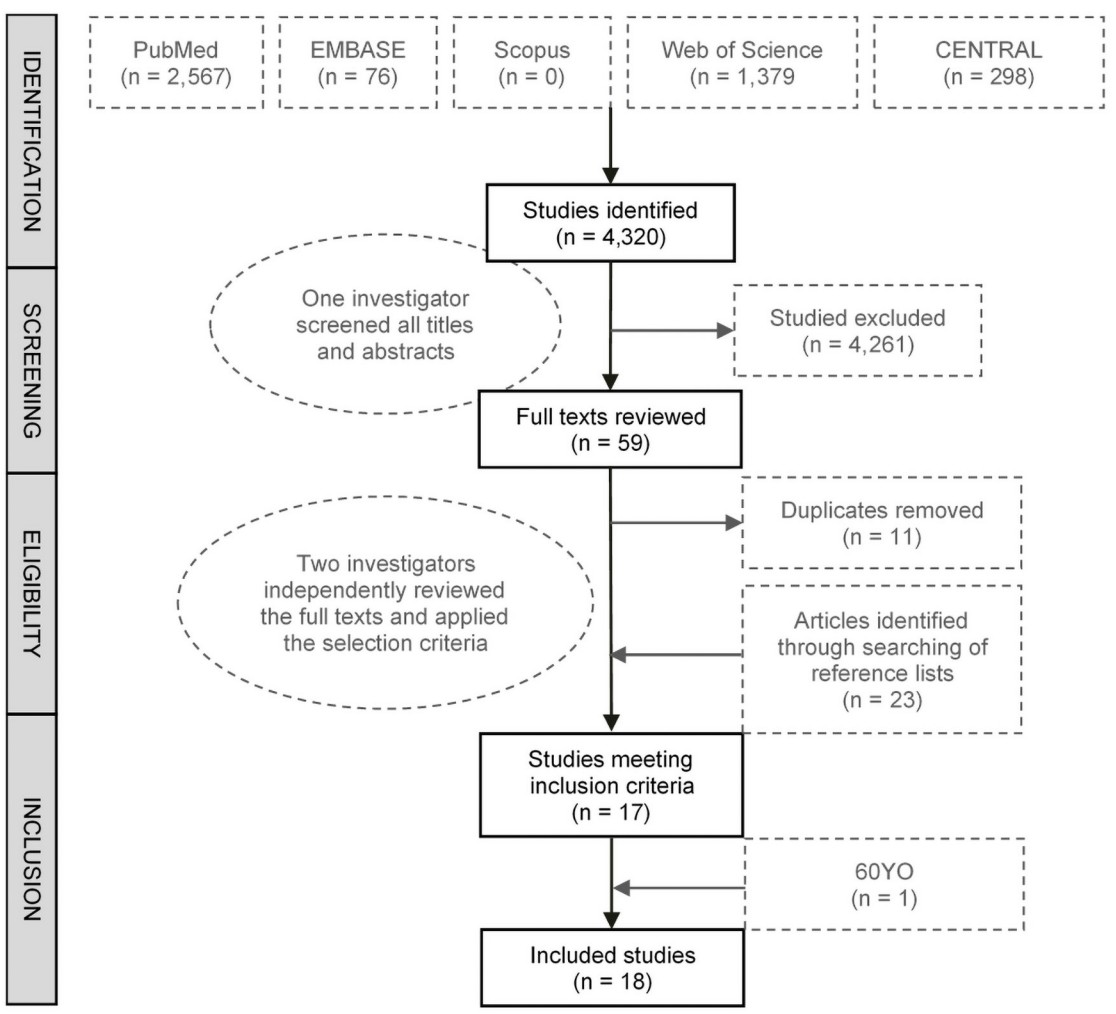

**Fig 3. Flow chart of systematic review and selection process.**

with all-cause mortality compared to low levels. These findings appear consistent with those of Iggman and colleagues, who examined adipose tissue 15:0 and 17:0 and also did not detect a significant association with all-cause mortality [39]. Our findings therefore do not support the contention that dairy fat intake, even at the high levels in Nordic countries, might contribute to higher risk of all-cause mortality. However, our systematic search identified relatively few studies that have evaluated dairy fat biomarkers and all-cause mortality, highlighting the need for more studies.

Our systematic review builds on and substantially extends previous meta-analyses [21,52]. Our results confirm the favourable cardiovascular benefits of having higher levels of 17:0 [21,52]. Our synthesis of the literature further generated novel evidence relating higher 15:0 levels with lower CVD risk. In comparison with previous meta-analyses that did not find evidence of an association between 15:0 and CVD outcomes, our review had considerably greater statistical power by including more studies and cases ($n = 17$ versus n $\leq$ 12, with approximately 5 times the number of incident CVD cases), consistently used study effect estimates from models adjusting for key potential confounders, and included studies from more diverse countries and geographies. The observed association of 15:0 with total CVD further supports

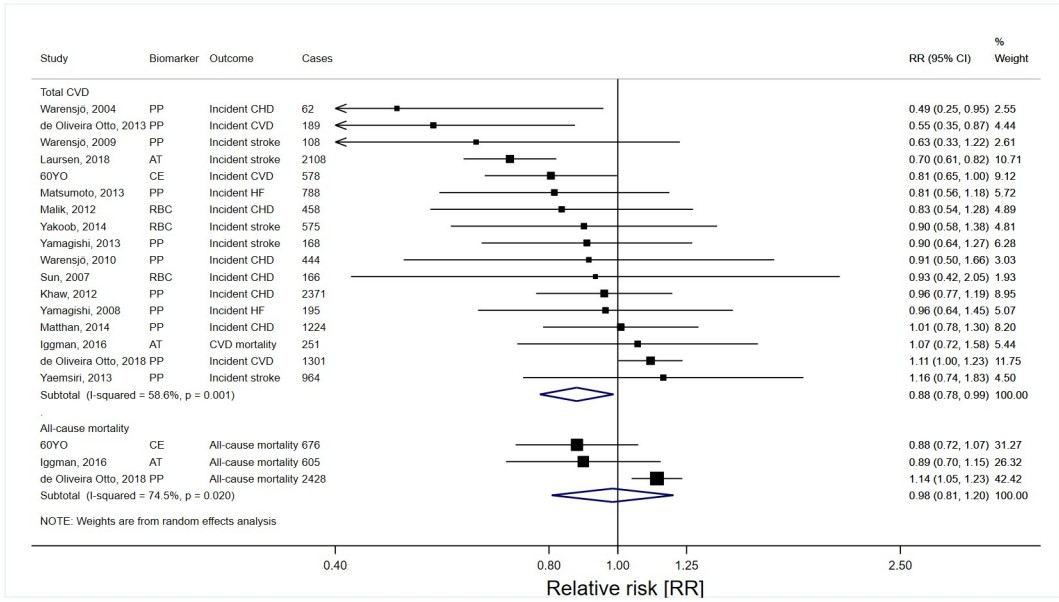

**Fig 4. Risk estimates for CVD incidence and all-cause mortality in the top tertile of pentadecanoic acid (15:0) relative to the bottom tertile.** AT, adipose tissue; CE, cholesterol ester; CHD, coronary heart disease; CI, confidence interval; CVD, cardiovascular disease; HF, heart failure; PP, plasma phospholipids; RBC, red blood cell (erythrocyte); RR, relative risk.

the inverse relationship between dairy fat intake and CVD risk. Interestingly, our findings suggest a null association between $t$16:1n-7 and CVD risk, consistent with earlier reviews [21]. However, $t$16:1n-7 is intercorrelated with the 15:0 and 17:0, and each of these dairy fat biomarkers was associated with lower risk of type 2 diabetes, which is a major risk factor of CVD

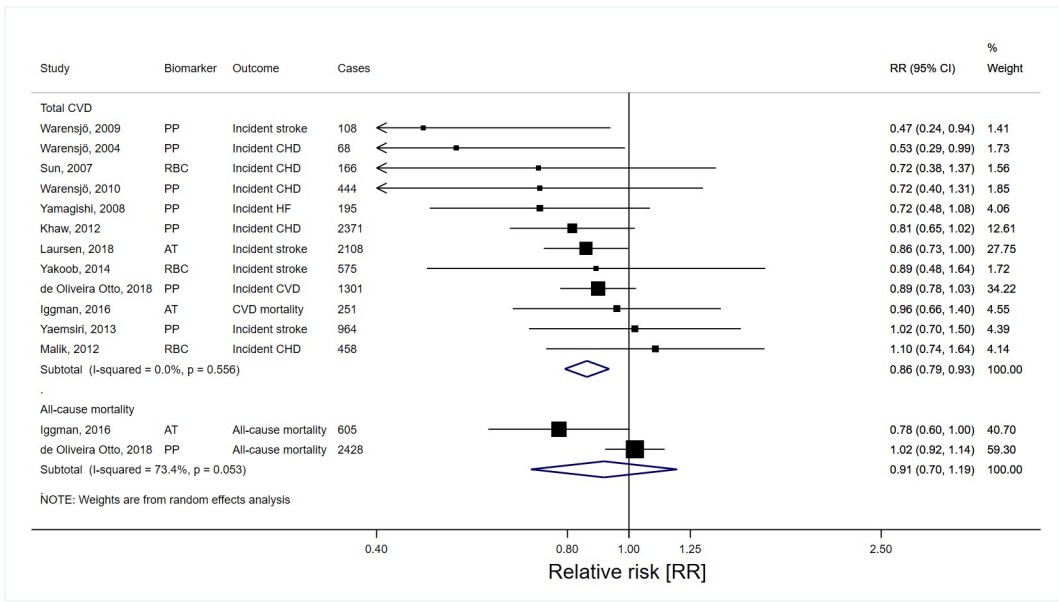

**Fig 5. Risk estimates for CVD incidence and all-cause mortality in the top tertile of heptadecanoic acid (17:0) relative to the bottom tertile.** AT, adipose tissue; CE, cholesterol ester; CHD, coronary heart disease; CI, confidence interval; CVD, cardiovascular disease; HF, heart failure; PP, plasma phospholipids; RBC, red blood cell (erythrocyte); RR, relative risk.

**Table 4. Pooled risk estimates of CVD subtypes and all-cause mortality in the top versus bottom tertiles of 15:0, 17:0, and $t$16:1n-7[1].**

| | Studies (n) | Cases (n) | Risk estimate (95% CI) | $I^2$ (%) |
|---|---|---|---|---|
| **15:0** | | | | |
| Total CVD | 17 | 11,950 | 0.88 (0.78, 0.99) | 58.6 |
| CVD incidence | 3 | 2,068 | 0.84 (0.59, 1.18) | 86.0 |
| CVD mortality | 3 | 1,282 | 1.10 (0.97, 1.25) | 0.0 |
| CHD incidence | 9 | 6,133 | 0.88 (0.75, 1.03) | 54.3 |
| CHD mortality | 1 | 567 | 1.14 (0.95, 1.35) | - |
| Stroke incidence | 7 | 4,644 | 0.88 (0.73, 1.07) | 64.8 |
| Stroke mortality | 1 | 188 | 1.09 (0.79, 1.51) | - |
| HF incidence | 2 | 983 | 0.88 (0.66, 1.16) | 0.0 |
| All-cause mortality | 3 | 3,709 | 0.98 (0.81, 1.20) | 74.5 |
| **17:0** | | | | |
| Total CVD | 12 | 9,009 | 0.86 (0.79, 0.93) | 0.0 |
| CVD incidence | 1 | 1,301 | 0.89 (0.78, 1.03) | - |
| CVD mortality | 2 | 1,084 | 0.84 (0.71, 1.00) | 0.0 |
| CHD incidence | 6 | 4,383 | 0.86 (0.78, 0.96) | 0.0 |
| CHD mortality | 1 | 567 | 0.85 (0.69, 1.05) | - |
| Stroke incidence | 5 | 4,284 | 0.87 (0.77, 0.98) | 0.0 |
| Stroke mortality | 1 | 188 | 0.63 (0.43, 0.93) | - |
| HF incidence | 1 | 195 | 0.72 (0.48, 1.08) | - |
| All-cause mortality | 2 | 3,003 | 0.91 (0.70, 1.19) | 73.4 |
| **$t$16:1n-7** | | | | |
| Total CVD | 6 | 3,477 | 1.01 (0.91, 1.12) | 0.0 |
| CVD incidence | 2 | 1,490 | 1.03 (0.91, 1.16) | 0.0 |
| CVD mortality | 1 | 833 | 1.02 (0.88, 1.19) | - |
| CHD incidence | 4 | 1,646 | 1.08 (0.95, 1.23) | 0.0 |
| CHD mortality | 1 | 567 | 1.09 (0.89, 1.33) | - |
| Stroke incidence | 2 | 1,104 | 1.10 (0.92, 1.33) | 0.0 |
| Stroke mortality | 1 | 188 | 0.85 (0.60, 1.22) | - |
| HF incidence | 1 | 788 | 0.81 (0.61, 1.08) | - |
| All-cause mortality | 1 | 2,428 | 1.07 (0.97, 1.17) | - |

[1]CHD, coronary heart disease; CI, confidence interval; CVD, cardiovascular disease; HF, heart failure.

[53,54]. The different association between the odd-chain fatty acids and $t$16:1n-7 could reflect true differences in their influence on cardiovascular health, or may be due to the relatively fewer number of studies ($n = 7$) that have investigated $t$16:1n-7 and/or potential greater measurement errors given the low levels and limited variance of circulating $t$16:1n-7.

Our study had several strengths. Firstly, the 60YO cohort is a large population-based prospective study with a high participation rate (78%), which reduces the risk of recall and selection biases and enhances the generalisability. Similarly, the inclusion of diverse populations of different age groups, sex, ethnicities, and countries in the systematic review enhances the generalisability of the findings. Secondly, the inclusion of >40,000 participants and >11,000 CVD events in the meta-analysis provides stronger statistical power than previous systematic reviews. Thirdly, dairy fat intake in the cohort study and meta-analysis of prospective studies was measured using objective biomarkers as opposed to a self-reported questionnaire, which avoids self-report or memory bias and errors from inaccurate nutrient composition

information, and better captures hidden dairy fat intake in mixed or prepared dishes [12]. Additionally, the biomarkers also allow for investigations into individual dairy fat biomarkers, which may have different biological effects. Lastly, the prospective design of 60YO cohort study and other studies included in the meta-analysis reduced the risk of recall and interviewer bias.

Some limitations of the 60YO cohort study were that serum 15:0 was measured once at baseline, which may have led to misclassification of exposure levels, although such misclassification is likely random and thus may have attenuated our results towards the null. The Swedish hospital discharge and deaths register is traditionally considered accurate, yet some deaths may be misclassified. We also cannot exclude residual confounding from inaccurately measured factors or factors not measured. The vast majority (89%) of the 60YO cohort were born in Sweden (81%) or Finland (8%), and extrapolation of the findings to other ethnic groups should be done with caution. Also, the studies included in the review were from the US, Sweden, Denmark, and the United Kingdom, which limits its generalisability to other regions. Despite several advantages of evaluating fatty acid biomarkers, the results cannot distinguish between different types of dairy foods (e.g., cheese, milk, butter, and yoghurt), which could have differential effects on health [2,55]. For example, butter intake increases total and LDL cholesterol when compared to cheese [56], and while cheese intake has been linked to lower risk of CVD outcomes [7,56–58], similar associations have not been reported for butter [57–59], which instead was recently linked to increased cardiovascular mortality in a large US cohort [60]. Additionally, the odd-chain saturated fats can be found at lower concentrations in other foods such as meat and fish and can potentially be produced endogenously [12,54]. However, these fatty acids primarily reflect the intake of dairy foods in most Western populations (as shown by the correlation between serum 15:0 and the dairy intake score in the 60YO Swedish cohort in Fig A in S1 File) [12], given the relatively high intake of dairy compared to fish, and, also, dairy (especially cheese) is the major dietary source of propionate, a primary substrate for the potential endogenous synthesis of odd-chain fatty acids [54]. It is unlikely that the inverse associations are confounded by meat intake, as meat is not associated with lower CVD risk [61]. Furthermore, we observed no change in the association between 15:0 and incident CVD after adjustment for the intake of vegetables, fruit and berries, fish, and meat in the Swedish cohort. In future studies, a detailed dietary assessment should be conducted to investigate and adjust for potential interrelationship between intake of dairy fat, total energy, and macronutrients such as carbohydrates. Our study-level meta-analysis has known limitations such as potential for increased heterogeneity due to differences in study design (e.g., covariate selection) and limited number of studies evaluating certain outcomes (e.g., all-cause mortality). We assumed log-linear associations of biomarkers with outcomes when transforming risk estimates to allow comparison of top versus bottom biomarker tertiles, which may over- or underestimate the estimates. However, meta-analysing risk estimates per biomarker SD in the subset of studies with relevant information available or retrieved from study authors provided similar results. Finally, the limited number of studies per lipid fraction, fatty acid, and outcome combination prevented us from evaluating nonlinear associations using dose–response meta-regression. Many of the limitations to our meta-analysis could be addressed by de novo individual-level pooled analyses of prospective studies utilising harmonised analysis protocols with predefined exposures, outcomes, and models [54,62].

## Conclusions

Higher circulating pentadecanoic acid (15:0), a biomarker of dairy fat intake, was associated with lower risk of CVD in this large population-based cohort study in Sweden. Our meta-analysis supports this finding, showing that higher levels of both odd-chain dairy fat biomarkers

15:0 and 17:0 were associated with lower CVD risk but not *t*16:1n-7. Our findings call for clinical and experimental studies to ascertain the causality of the relationship and the potential role of dairy foods in CVD prevention.

## Supporting information

**S1 Checklist. STROBE statement for the reporting of cohort studies.**
(DOCX)

**S2 Checklist. PRISMA checklist.**
(DOCX)

**S1 Protocol. Prespecified analytical plan for 60YO cohort study.**
(DOCX)

**S1 Text. Health screening and questionnaire.**
(DOCX)

**S2 Text. Search strategy and methods.**
(DOCX)

**S1 File. Supporting information tables and figures.** Table A. 60YO study population characteristics at baseline by quartile of serum cholesterol ester pentadecanoic acid (15:0). Table B. Hazard ratios of primary (incident CVD and all-cause mortality) and secondary outcomes (incident CHD, stroke, and CVD mortality) with serum 15:0 evaluated per interquintile range, per SD, or per % of totals fatty acids in the 60YO study. Table C. Hazard ratios of incident CHD, stroke, and CVD mortality per interquintile range of serum pentadecanoic acid (15:0) in the 60YO study. Table D. Hazard ratios (95% CI) of incident CVD and all-cause mortality by serum pentadecanoic acid (15:0) (per 1 IQR increase) according to sex, BMI, and serum proportions of long-chain n-3 PUFA in the 60YO study. Table E. Hazard ratios (95% CI) of incident CVD and all-cause mortality by serum pentadecanoic acid (15:0) (per 1 IQR increase) assessed in sensitivity analyses excluding early cases, censoring at 10 years of follow-up or by excluding individuals with prevalent CVD at baseline in the 60YO study. Table F. Characteristics of studies included in the systematic review. Table G. Newcastle–Ottawa Score (NOS) calculation for studies included in the systematic review. Table H. Pooled risk estimates of cardiovascular disease (CVD) subtypes and all-cause mortality per standard deviation (SD) increase in 15:0, 17:0, and *t*16:1n-7. Table I. Risk estimates of total cardiovascular disease (CVD) comparing top versus bottom tertile of 15:0, 17:0, and t16:1n-7 in subgroups by age, sex, duration of follow-up, or study location. Fig A. Relationship between the dairy intake score and pentadecanoic acid in serum cholesterol esters, evaluated using restricted cubic splines and adjusted for age, sex, BMI, physical activity, alcohol use, and smoking status in the 60YO study. The circles represent the point estimates and the error bars, 95% CIs. The dairy intake score was based on self-reported habits regarding use of butter, cheese, milk, and yoghurt [1]. The histogram shows the distribution of the dairy intake score in the cohort. Fig B. Funnel plot of studies included in the meta-analysis for serum 15:0 (A), 17:0 (B), and t16:1n-7 (C).
(DOCX)

## Acknowledgments

The authors thank Siv Tengblad for assessment of FA composition and all cohort participants for their contributions.

## Author Contributions

**Conceptualization:** Jason H. Y. Wu.

**Data curation:** Saiuj Bhat, Zhaoli Dai.

**Formal analysis:** Saiuj Bhat, Jason H. Y. Wu, Matti Marklund.

**Methodology:** Kathy Trieu.

**Supervision:** Jason H. Y. Wu, Ulf Risérus, Matti Marklund.

**Writing – original draft:** Kathy Trieu.

**Writing – review & editing:** Kathy Trieu, Saiuj Bhat, Zhaoli Dai, Karin Leander, Bruna Gigante, Frank Qian, Andres V. Ardisson Korat, Qi Sun, Xiong-Fei Pan, Federica Laguzzi, Tommy Cederholm, Ulf de Faire, Mai-Lis Hellénius, Jason H. Y. Wu, Ulf Risérus, Matti Marklund.

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
