## [Editor Report · Decision Letter 0]

18 Feb 2021

Dear Dr Marklund, 

Thank you for submitting your manuscript entitled "Biomarkers of dairy fat intake, incident cardiovascular disease, and all-cause mortality: a cohort study, systematic review, and meta-analysis" for consideration by PLOS Medicine.

Your manuscript has now been evaluated by the PLOS Medicine editorial staff as well as by an academic editor with relevant expertise and I am writing to let you know that we would like to send your submission out for external peer review.

Kind regards,

Dr Raffaella Bosurgi

Executive Editor

PLOS Medicine

---

## [Decision Letter · Decision Letter 1]

24 Jun 2021

Dear Dr. Marklund,

Thank you very much for submitting your manuscript "Biomarkers of dairy fat intake, incident cardiovascular disease, and all-cause mortality: a cohort study, systematic review, and meta-analysis" (PMEDICINE-D-21-00856R1) for consideration at PLOS Medicine. We apologize for the delay in sending you a response. 

Your paper was discussed with an academic editor with relevant expertise and sent to independent reviewers, including a statistical reviewer. The reviews are appended at the bottom of this email and any accompanying reviewer attachments can be seen via the link below:

[LINK]

In light of these reviews, we will not be able to accept the manuscript for publication in the journal in its current form, but we would like to invite you to submit a revised version that addresses the reviewers' and editors' comments fully. You will appreciate that we cannot make a decision about publication until we have seen the revised manuscript and your response, and we expect to seek re-review by one or more of the reviewers. 

We hope to receive your revised manuscript by Jul 15 2021 11:59PM. Please email us (plosmedicine@plos.org) if you have any questions or concerns.

Please let me know if you have any questions, and we look forward to receiving your revised manuscript. 

Sincerely,

Richard Turner, PhD

rturner@plos.org

Noting PLOS' data policy, https://journals.plos.org/plosmedicine/s/data-availability, please adapt your data statement (submission form) so that the point of contact for inquiries about data is not an author, and preferably not an individual. 

At the start of the abstract, we suggest adding an introductory sentence, say, to state the study's aim.

Please quote aggregate demographic details for participants in the cohort study in your abstract.

Please adapt your abstract by adding a new final sentence to the "Methods and findings" subsection, which should begin "Study limitations include ..." or similar and should quote 2-3 of the study's main limitations. 

After the abstract, please add a new and accessible Author Summary section in non-identical prose. You may find it helpful to consult one or two recent research papers in PLOS Medicine to get a sense of the preferred style. 

Early in the Methods section of your main text, please state whether or not the study had a protocol or prespecified analysis plan, and if so attach the relevant document(s) as a supplementary file(s), referred to in the text. 

Please highlight analyses that were not prespecified. 

Are you able to update your search so as to include additional studies in the analysis?

Throughout the text, please adapt reference call-outs to the following style: "... processing methods [4,5]." (noting the absence of spaces within the square brackets). 

Please remove the information on funding, competing interests and data access from the end of the main text. In the event of publication, this information will appear in the article metadata, via entries in the submission form. Ethics information should be moved to the Methods section. 

Please review the reference list to ensure that all entries comply with journal style. Italics should be converted to plain text. Six author names should be listed rather than 3, followed by "et al.", where appropriate.

Are references 17 & 48 missing journal names?

Please use the journal name abbreviations "PLoS ONE" and "PLoS Med.".

Please add completed STROBE (for the observational element of your study) and PRISMA (for the meta-analysis) checklists as supplementary files, labelled "S1_STROBE_Checklist" or similar and referred to as such in the Methods section. 

In the checklists, please refer to individual items by section name, e.g., "Methods" and paragraph number, not by line or page numbers as these generally change in the event of publication.

Comments from the reviewers:

*** Reviewer #1: 

This study aims to investigate the association of serum pentadecanoic acid (15:0) with incident CVD and all-cause mortality in a Swedish cohort study, as well as systematically review studies of the association of dairy fat biomarkers with CVD outcomes or all-cause mortality. 

Comments:

This article presents itself as two studies in one, which is not necessarily a bad thing (although may create potential for publication bias, if one of the research pieces is stronger that the other per se). 

Did the authors consider splitting this into two separate papers for publication?

"The Stockholm Cohort of 60-year-olds (60YO) has been previously described. Of 5,460 randomly selected individuals invited, 4,232 (78%) agreed to participate (52% women) and provided informed consent. For the current analysis, 4,150 participants that had fasting blood samples collected at baseline between 1997 and 1999, and follow-up information until December 31, 2014 were included."

Can the authors please comment on whether the 4150 included participants are representative of the wider population?

"Cox proportional hazard models were used to estimate hazard ratios (HRs) and 95% confidence intervals (CIs) for the association between serum 15:0 with primary and secondary outcomes. Differences in time to first CVD event or death by serum 15:0 levels were estimated using Laplace regression."

Technically appropriate statistical models and methods have been applied by the authors.

"During follow-up, around 15% of participants died and a similar number of persons experienced a CVD event. Hence, we estimated the 15th percentile difference (PD) defined as the difference in time (months) by which 15% of exposed vs unexposed had died or experienced an incident CVD event. "

This is an interesting approach. Did the authors consider performing this analysis for a variety of percentiles and plotting these against time difference to visualise the trend of percentile differences?

"Three models were evaluated: 1) crude, without adjustments; 2) age- and sex adjusted; and 3) multivariable-adjusted including age, sex, BMI, smoking, physical activity, education, alcohol intake, diabetes, drug-treated hypertension and drug-treated hypercholesterolaemia as covariates".

This is a thorough and robust approach to take for the modelling. Did the authors consider including ethnicity or race as a covariate in the adjusted models?

"Multiple imputations (n=20) were conducted to account for missing covariates."

Again, the authors have opted for a suitable and rigorous analytical approach here to account for missing data. 

Did the authors complete any sensitivity analyses on this, perhaps by running the analysis with missing data excluded?

How much missing data was there? 

"Serum 15:0 was assessed as a continuous variable (per interquintile range, IQR), defined as the range between the 90th and 10th percentiles) or a categorical variable (quartiles)."

The authors have demonstrated good practice by considering two ways of treating this variable within the modelling. Please note the typo of a surplus parenthesis.

"There was no violation of the proportional hazard assumption based on visual examination of Schoenfield residuals. Restricted cubic splines were used to evaluate potential nonlinear associations".

The authors have appropriately checked model assumptions for validity.

"We explored the associations between serum 15:0 and outcomes in subgroup analyses stratified by sex, BMI and serum n-3 PUFA subgroups (< median vs ≥ median)."

The subgroup analyses completed by the authors provide great insight to the data and patterns within. Did they additionally consider investigating age as a subgroup analysis?

"We conducted sensitivity analyses by 1) adjusting for self-reported dietary habits (vegetable, fruit and berries, lean fish, oily fish, and processed meat intake), 2) excluding participants with prevalent CVD also from analyses of all-cause mortality (in line with analyses of CVD outcomes), 3) restricting analyses to the first 10 years of follow-up to minimise misclassifications attributable to exposure changes over time, and 4) excluding cases in the first two years of follow-up to avoid reverse causation because of undetected disease or presence of risk factors at baseline."

A thorough array of sensitivity analyses have been completed by the authors.

"The systematic review followed the PRISMA guidelines and was registered on PROSPERO 8 [CRD42020162551]. "

Can the authors please provide the PRISMA checklist and protocol within the supplementary material?

"Two reviewers (SB and ZD) independently screened the studies for eligibility, extracted data, and assessed the quality of studies using the Newcastle-Ottawa Scale (NOS). "

Can the authors please clarify here how cases of disagreement, if any, were settled? Was this by consensus or by an independent third reviewer, perhaps?

"Pooled associations of 15:0, 17:0, and t16:1n-7 with CVD outcomes and all-cause mortality were estimated using random effects meta-analysis."

The authors have applied a suitable modelling approach that can account for heterogeneity between the studies.

"Risk estimates (HR, odds ratio, or relative risk, RR) for each study were transformed to allow consistent comparisons between the top and bottom tertile of fatty acid distributions (Supplementary File 2)"

Can the authors please note in the limitations the risk of over/under stating these estimates by transforming them?

"The I 2 and Q statistics were used to assess heterogeneity of included studies. Publication bias was assessed by visual inspection of funnel plots and statistically using Egger's and Begg's tests. Stratified meta-analyses were performed on subgroups defined by age, sex, follow-up duration, and geographic region. We repeated the meta-analysis using a fixed effects models in a sensitivity analysis."

The authors have demonstrated that they have followed a robust and rigorous analytical approach.

Furthermore, a thorough and informative selection of tables and figures have been presented within the manuscript.

*** Reviewer #2: 

The study 'Biomarkers of dairy fat intake, incident cardiovascular disease, and all-cause mortality: a cohort study, systematic review, and meta-analysis' investigated the association of serum pentadecanoic acid (15:0), a biomarker of dairy fat intake, with incident CVD and all-cause mortality. The manuscript is well written, the analyses and conclusions are sound. The topic is of importance and of interest; it is commendable that the authors additionally present a meta-analysis in support of their findings.

A few suggestions:

1. Can the authors correlate serum cholesterol ester pentadecanoic acid (15:0) with a measure of diet quality (e.g. Healthy Eating Index, alternate Healthy Eating Index or DASH diet quality index) in the Stockholm cohort.

2. Sensitivity Analysis: Please additionally adjust your analysis for 'Total Energy Intake'.

3. Supplemental material: Please additionally explain how covariates were assessed in the Stockholm cohort. Also, please add a p-value to Suppemental Table 3 to assess if there were significant differences across quartiles.

4. If available, please add serum lipid level data (LDL, Tg, total Chol, HDL) according to quartiles of serum cholesterol ester pentadecanoic acid (15:0) to the results. 

5. If individuals consume more dairy fat (i.e. these individuals will have increased levels of serum cholesterol ester pentadecanoic acid (15:0)), they in turn will consume less amount of other macronutrients (in particular carbs). Can the authors show data to this respect ? Second, does carbohydrate quality modify serum levels of cholesterol ester pentadecanoic acid (15:0) ?

6. Sensitivity analysis, please additionally adjust your analysis for 'carbohydrate intake'.

*** Reviewer #3: 

The article "Biomarkers of dairy fat intake, incident cardiovascular disease, and all-cause mortality: a cohort study, systematic review, and meta-analyses" authored by Trieu et al. has been reviewed. Overall, this study provides additional evidence that markers of dairy fat intake seem to be associated with favorable cardiovascular outcomes and that there is no association between dairy intake and mortality. Indeed, the unusual but additional inclusion of the systematic review and meta-analyses supports the findings of the authors and strengthens the underlying message of the paper. However, I have some queries and suggestion below that are important to the manuscript. Good luck. 

Comments:

Methods: Page 5 Line 10 - granted you have stated that measurement were only analyzed at baseline, but were follow-up plasma measurements taken? Was dairy intake and fatty acid composition assessed during follow-up to see if these measurements had changed much over time?

Methods: Page 5 Line 15 - please detail the gas chromatography methods and isolation of serum cholesterol ester. 

Why choose cholesterol esters. Phospholipids also contain 15:0 and 17:0. I assume it is partially because a blood sample is easier to get then other tissues. As a matter of interest, is s 15:0 and 17:0 measurable in RBCs?

What were exclusion criteria? Was BMI one of them, it seems unusual that the BMI range in table 1 is so low and that BMI > 33 was not reported in 60yo cohort. Indeed, obesity was also not reported in table 1. Certainly in 2020 over 54% of Swedish individuals over the age 60 have a higher BMI than 25

Move supplementary file 8 to the methods section

Please provide figure legends for all figures. It is difficult to follow figures 1 and 2.

Introduction lines 2-14 - probably worth mentioning as an addition to the point made on line 9 that the evidence seems to suggest that fermented dairy products (yoghurt/cheese) in particular may be more protective than milk alone (https://doi.org/10.3390/foods7030029;
https://doi.org/10.1093/advances/nmz069).

Editor comments: 

Overall, I am supportive of the publication of this manuscript, but I have suggested revisions that require attention.

***

[LINK]

---

## [Decision Letter · Decision Letter 2]

30 Jul 2021

Dear Dr. Marklund,

Thank you very much for re-submitting your manuscript "Biomarkers of dairy fat intake, incident cardiovascular disease, and all-cause mortality: a cohort study, systematic review, and meta-analysis" (PMEDICINE-D-21-00856R2) for consideration at PLOS Medicine.

I have discussed the paper with our academic editor and it was also seen again by three reviewers. I am pleased to tell you that, provided the remaining editorial and production issues are fully dealt with, we expect to be able to accept the paper for publication in the journal.

[LINK]

Please let me know if you have any questions, and we look forward to receiving the revised manuscript shortly.   

Sincerely,

Richard Turner, PhD

rturner@plos.org

Requests from Editors:

If available, please add a web address for the Karolinska Institute to your data statement. 

Rather than "60 year olds" in the abstract, please quote "... median age 60.5 years" or similar. 

Where you quote measures of risk in the abstract, please also quote p values as in the main text.

Please adapt the wording in the "Conclusions" subsection of your abstract to note that you have also done a new cohort study. 

In the abstract and text you mention that "prospective" studies were included in the meta-analysis, and we ask you to remove this word from the abstract. Although we are aware that views differ, judging from the description of the research designs included in the main text we doubt that all these studies meet the definition of a prospective study that we generally use at PLOS Medicine.

In the author summary, we suggest "it is important ..." rather than "it is critical ...".

Please adapt the reference call-outs throughout the ms to the following style (noting the absence of spaces within the square brackets): " ... [4,6,7]. For example ...".

In the reference list, please remove the information on competing interests from references 3 & 61, and any other relevant citations. 

Comments from Reviewers:

*** Reviewer #1: 

The authors have satisfactorily responded to each comment in turn, amending the manuscript accordingly. 

*** Reviewer #2: 

The authors responded well to my comments. 

*** Reviewer #3: 

I have no further comments to add. Well done on your excellent paper and good luck with your research.

***

[LINK]

---

## [Editor Report · Decision Letter 3]

11 Aug 2021

Dear Dr Marklund, 

On behalf of my colleagues and the Academic Editor, Dr Basu, I am pleased to inform you that we have agreed to publish your manuscript "Biomarkers of dairy fat intake, incident cardiovascular disease, and all-cause mortality: a cohort study, systematic review, and meta-analysis" (PMEDICINE-D-21-00856R3) in PLOS Medicine.

Prior to final acceptance, please adapt the wording of the data statement to "Requests from researchers interested in accessing study data can be sent to the Karolinska Institutet ..." or similar; and split the final summary point into two (e.g., the second could begin "Further trials are needed to explore if and how ...").

Please also adapt all reference call-outs so that they precede items of punctuation (e.g., "... in a large US cohort [62].").

PRESS

Sincerely, 

Richard Turner, PhD 

rturner@plos.org